# Detailed mapping of *Bifidobacterium* strain transmission from mother to infant via a dual culture-based and metagenomic approach

Conor Feehily [1,2,3,12], Ian J. O'Neill [2,4,12], Calum J. Walsh[1,2,12], Rebecca L. Moore[5], Sarah Louise Killeen[5], Aisling A. Geraghty[5], Elaine M. Lawton[1,2], David Byrne[5], Rocio Sanchez-Gallardo [2,4], Sai Ravi Chandra Nori [1,2,6], Ida Busch Nielsen[2,4], Esther Wortmann[2,4], Elizabeth Matthews[7], Roisin O'Flaherty [7,8], Pauline M. Rudd[7,9], David Groeger[10], Fergus Shanahan [2], Radka Saldova [7,11], Fionnuala M. McAuliffe[5,13], Douwe Van Sinderen [2,4,13] ✉ & Paul D. Cotter [1,2,13]

A significant proportion of the infant gut microbiome is considered to be acquired from the mother during and after birth. Thus begins a lifelong and dynamic relationship with microbes that has an enduring impact on host health. Based on a cohort of 135 mother-infant (F = 72, M = 63) dyads (MicrobeMom: ISRCTN53023014), we investigated the phenomenon of microbial strain transfer, with a particular emphasis on the use of a combined metagenomic-culture-based approach to determine the frequency of strain transfer involving members of the genus *Bifidobacterium*, including species/ strains present at low relative abundance. From the isolation and genome sequencing of over 449 bifidobacterial strains, we validate and augment metagenomics-based evidence to reveal strain transfer in almost 50% of dyads. Factors important in strain transfer include vaginal birth, spontaneous rupture of amniotic membranes, and avoidance of intrapartum antibiotics. Importantly, we reveal that several transfer events are uniquely detected employing either cultivation or metagenomic sequencing, highlighting the requirement for a dual approach to obtain an in-depth insight into this transfer process.

The establishment of the gut microbiota in the perinatal period is a key event in early life. The microbes that colonise the infant's digestive tract are, among other functions, essential for educating the immune system[1–3]. Perturbations in the gut microbiota composition in early life, for example as a result of antibiotic treatment, are also associated with gastrointestinal infection, atopic disease and obesity[4,5].

Concurrent with birth, the neonatal gut becomes colonised by so-called founder species, many of which are representatives of the genera *Escherichia*, *Enterococcus*, and *Lactobacillus* (more recently reassigned as several distinct genera). This facilitates subsequent colonisation by strict anaerobes such as members of the *Bifidobacterium* and *Bacteroides* genera, which frequently dominate a healthy infant gut within 1–4 weeks[6–10].

Vertical transmission of bacteria, i.e., from mother to infant, is assumed to be key in the establishment of the infant gut microbiome[11,12]. Shotgun metagenomic analyses can infer strain transmission, either by aligning species-specific marker genes[7,8,13,14] or via alignment of metagenomic reads to reference genomes[15]. These methods have indicated that 50–80% of all species in the infant gut at 4-months are shared with the maternal microbiome. However, strain transmission rates vary between species[7,8], with

strains of *Bifidobacterium* and *Bacteroides* being most frequently transmitted[7,8,14].

Strains that are vertically transmitted appear to be more persistent in the infant than non-vertically transmitted strains, highlighting the importance of vertical transmission to microbiota formation and stability[14]. Successful transmission is impacted by antibiotic use in caesarean section as, for example, transmission of *Bifidobacterium longum* subsp. *longum* appears to be infrequent and delayed in infants born by caesarean birth[16]. Furthermore, transmission of certain *Bacteroides* species is almost absent in infants born by caesarean birth[7] and the prevalence of *Staphylococcus*, *Klebsiella*, *Escherichia*, and *Clostridium* in infants born by caesarean birth can increase vulnerability to gastrointestinal infection[17,18]. However, it remains unclear if other maternal factors, such as diet and health can influence the infant's microbiome development, or if modulation of these factors via ingestion of specific probiotic strains influence strain transmission[19].

Here, we investigate the phenomenon of strain transfer employing a combination of high resolution metagenomic sequencing and direct culture isolations. We show that the use of a complementary culture-based approach reveals strains that are present at a level below the threshold at which detection through metagenomics alone was possible. In addition, by using this combined approach we show that delivery mode and exposure to antibiotics in labour are important factors that influence transmission.

## Results

### The infant and maternal gut microbiota are defined by different *Bifidobacterium* species

The samples analysed in this study were collected during the MicrobeMom study (ISRCTN53023014[20,21]), these included samples from the maternal stool, vagina, oral cavity, and breast milk in addition to infant stool (Table 1; Fig. 1a). From the 160 participants recruited, there were 135 dyads from which we received samples. Out of these, a total of 1012 high quality metagenomic samples were analysed post quality filtering, corresponding to 132 mother/infant dyads (Fig. 1a, Table 1, Supplementary Data 1).

Maternal stool samples yielded a median of 5.1 million high quality reads and were dominated by *Bacteroides* species throughout pregnancy (Supplementary Fig. S1a). *Bifidobacterium* spp. were also present across these samples, representing between 5 and 6% relative abundance of the stool microbiota (Fig. 1b). The two most dominant *Bifidobacterium* species were *Bifidobacterium longum* subsp. *longum* and *Bifidobacterium adolescentis*, with a mean relative abundance of 2.09% and 2.42%, respectively, and detected in 84.48% (311/368) and 68.63% (253/368) of maternal stool samples, respectively (Fig. 1b).

Where possible, DNA was extracted from the meconium sample to obtain an as close to baseline infant sample as possible. If not feasible, a stool sample collected within one-week following birth was analysed. In total, 53 meconium samples, 44 one-week samples, and 118 one-month infant stool samples were sequenced, generating an overall median of 4.4 million high quality reads, and analysed (Fig. 1a). Early (delivery or one-week) infant samples had a significantly (ANOVA $p \le 0.05$) lower alpha diversity compared to both maternal stool and infant one-month stool (Fig. 1c and Supplementary Fig. S1b). Overall, the relative abundance of *Bifidobacterium* increased with time in the infant stool samples, reaching a mean relative abundance of 52.04% at one-month, compared to 19.19% at delivery (Fig. 1b). A reflective increase in prevalence of *Bifidobacterium* was also observed, in 55.56% of delivery samples and 88.24% of one-month samples (Fig. 1b). When compared to maternal stool, both the relative abundance and prevalence levels of *Bifidobacterium breve* were higher in infant stool, while *Bifidobacterium bifidum* had a higher relative abundance in infant stool, but a lower overall prevalence across samples (Fig. 1b).

The oral microbiome of mothers at 16-weeks of pregnancy, as revealed by a median of 760 K high quality reads, was dominated by *Veillonella*, *Prevotella*, and *Streptococcus mitis* with only 2 samples (1.56%) containing *Bifidobacterium* at detectable levels (Supplementary Fig. S1a). Both the stool and oral microbiome had a comparably high α-diversity of species (Fig. 1c). The α-diversity of the vaginal microbiome (median of 396 K high quality reads) was much lower with a homogenous population dominated by *Lactobacillus crispatus*, *Lactobacillus iners*, *Lactobacillus gasseri* or *Lactobacillus jensenii* (Fig. 1c and Supplementary Fig. S1a). A *Bifidobacterium* species was detected in the vaginal microbiome of six individuals from at least one timepoint, with *B. breve* representing >40% relative abundance in 4 samples. Only 6.25% (8/128) of mothers were shown to possess a relatively diverse vaginal microbiome without dominance of *Lactobacillus* at any stage in their pregnancy. The vaginal community of individuals remained stable throughout the study.

Within breast milk samples (median of 685 K high quality reads), *Staphylococcus epidermidis* and *Streptococcus mitis* represented the most dominant taxa, with members of the genus *Bifidobacterium* also common, being present in 17.67% (12/68) of the samples, with *B. breve* the most abundant species from this genus (Supplementary Fig. S1a).

## Table 1 | Maternal and infant summary statistics

| | Total (*n* = 135) | |
|---|---|---|
| | *n*[a] | Value |
| Age (mean years, SD) | 133 | 33.53, 3.91 |
| Body mass index (median kg/m², IQR) | 135 | 24.35, 4.11 |
| Ethnicity (*n*, % White) | 129 | 96.96 |
| Education (*n*, % completed third level) | 134 | 85.80 |
| Multiparous (%) | 134 | 43.28 |
| Antibiotic use (%) | | |
| - At labour (maternal) | 135 | 33.33 |
| - In NICU (infant) | 135 | 6.67 |
| - At one-month visit (maternal) | 135 | 23.31 |
| - At one-month (infant) | 135 | 6.77 |
| Breastfeeding at 1 month (%) | | |
| - Breastmilk only | 126 | 48.41 |
| - Formula only | 126 | 19.84 |
| - Breastmilk and formula | 126 | 31.75 |
| Labour onset (%) | | |
| - Spontaneous | 135 | 57.78 |
| - Induced | 135 | 28.89 |
| - Pre-labour Caesarean birth | 135 | 12.59 |
| - Emergency Lower Segment Caesarean | 135 | 0.74 |
| Rupture of membranes (%) | | |
| - POM | 135 | 24.71 |
| - SROM | 135 | 42.22 |
| - At LSCS | 135 | 13.33 |
| - ARM | 135 | 16.30 |
| - SROM and ARM | 135 | 0.74 |
| Gestation in days (mean, SD) | 135 | 280.60, 9.00 |
| Infant sex (% female) | 135 | 53.33 |
| Birthweight (mean g, SD) | 135 | 3646.79, 525.58 |
| Delivery mode (% Vaginal) | 135 | 79.26 |
| Maternal Lewis status (% positive) | 81 | 93.83 |
| Maternal Secretor status (% positive) | 79 | 74.68 |

*POM* Puncture of Membranes, *SROM* Spontaneous Rupture of Membranes, *At LSCS* At Lower Segment Caesarean Section, *ARM* Assisted Rupture of Membrane.
[a]values in the *n* column represent the numbers to which data was available.

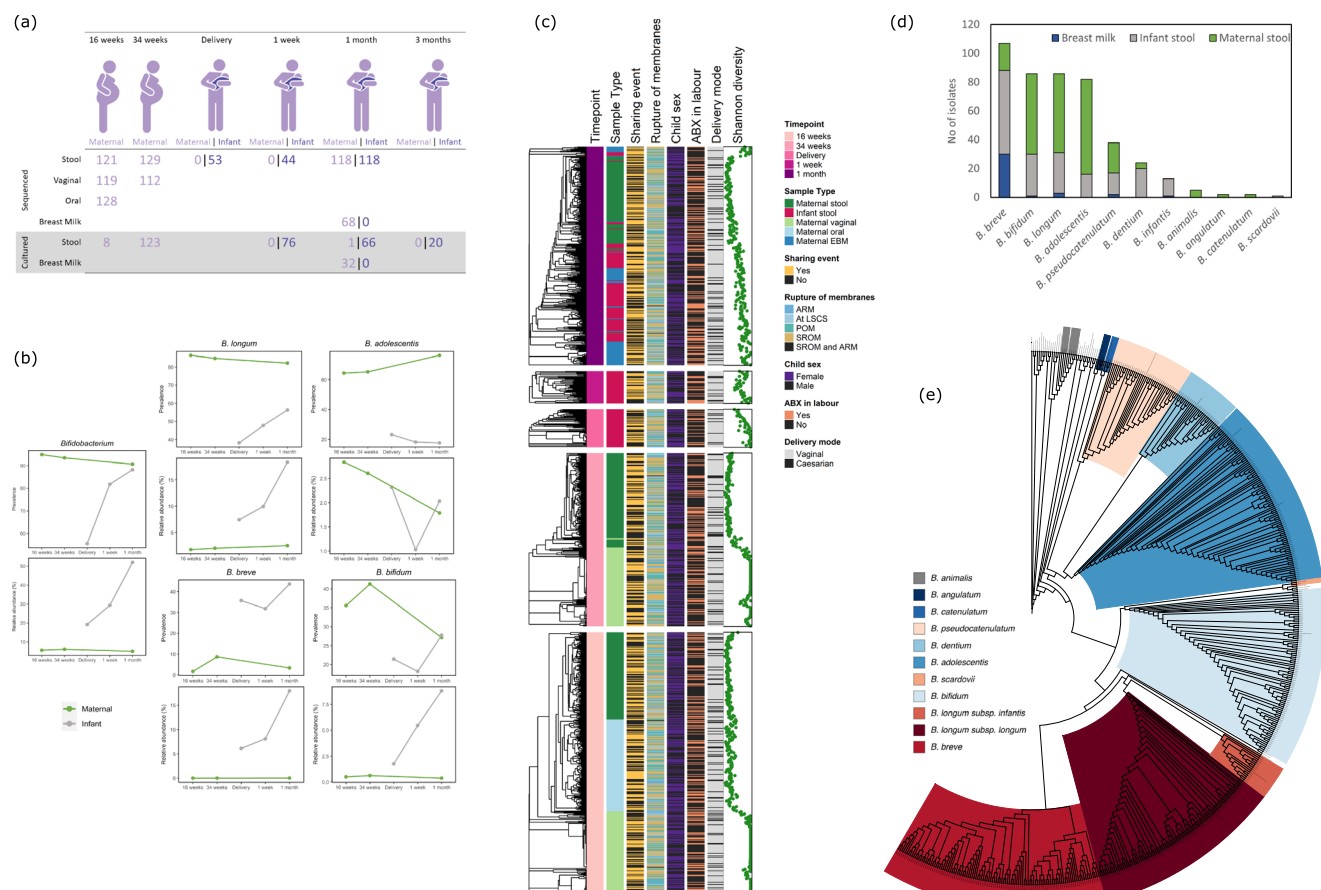

**Fig. 1 | Analysis of both the microbiota and cultured isolates in the study.**
**a** Overview of samples analysed by shotgun metagenomics (top white) and bifidobacterial culture isolations (bottom grey). Maternal samples are represented to the left of "|" whilst infant samples are on the right. The timepoint of sampling is shown across the top [Created with BioRender.com]. **b** The prevalence and relative abundance of the total *Bifidobacterium* genus in maternal ($n$ = 129, green) and infant ($n$ = 119, grey) stool as determined by metagenomic sequencing is represented in the left plots. Stratification for the 4 most dominant *Bifidobacterium* species is adjacent. **c** Clustering of the microbiome of all sequenced samples

($n$ = 1011) based on Bray-Curtis is shown by the dendrogram. Each block represents a different sampling timepoint as shown in the first coloured annotation bar. Further relevant covariates are annotated in subsequent colour bars. The alpha (Shannon) diversity of each sample is represented in the green dot plot. **d** Total number of culture-isolated and sequenced ($n$ = 489) *Bifidobacterium* species isolated from breast milk, maternal, and infant stool. **e** Phylogenetic tree of all *Bifidobacterium* strains isolated in this study together with type strains for all bifidobacterial species. Each coloured clade highlights the different species isolated.

## A limited number of covariates correlate with maternal or infant microbiota diversity

To investigate the factors that associate with the observed microbiota beta-diversity for each sample type, 22 different covariates were modelled with *envfit* from the vegan package[22] (Supplementary Fig. S1e). The infant stool microbiota at delivery was shown to be associated with maternal parity ($p$ = 0.002; Supplementary Fig. S1d). At 1-week, the infant microbiota was determined to be associated with maternal exposure to antibiotics in labour ($p$ = 0.003), while at 1-month the infant microbiota was observed to be significantly associated with the Lewis status of the mother ($p$ = 0.022). There were no significant associations observed between the infant microbiota at any timepoint and delivery mode or breast-feeding status.

Furthermore, the maternal stool microbiota at 1-month was shown to be significantly associated with delivery mode ($p$ = 0.025; Supplementary Fig. S1d), while the vaginal microbiota beta diversity was shown to be significantly associated with parity at both 16 and 34-weeks pregnancy ($p$ = 0.001 and 0.004, respectively; Supplementary Fig. S1d). Finally, breast milk microbiota diversity was determined to be associated with Lewis status ($p$ = 0.033), yet not with secretor status ($p$ = 0.710).

## Targeted isolation of *Bifidobacterium* strains reveals a high prevalence of *B. breve* in infant stool

To comprehensively assess *Bifidobacterium* strain transfer occurrence, and to complement and augment shotgun metagenomic data, isolation of *Bifidobacterium* was performed using selective media. In total, 133 mother/infant dyads were analysed including both stool and breast milk samples up to three-months post-partum (Fig. 1a, Supplementary Data 1). From these, *Bifidobacterium* was isolated from both mother and infants across 109 dyads, whilst a further 15 dyads provided a singleton case (i.e., isolation of bifidobacteria from only the mother or infant in each dyad; see Supplementary Data 1). A total of 510 individual colony isolates across all samples were purified and their genome sequenced. Following quality filtering, 449 bifidobacterial genomes with an average 365.60-fold genome coverage (353.31 median) were included for analysis (Supplementary Data 2). The median genome size was 2.32 Mb with an average of 25.82 contigs per genome and the genomic GC content range was 55.86–64.97% with a median of 59.41% in line with previous findings (Supplementary Fig. S1f[23];).

The most frequently isolated species was *B. breve* followed by *B. bifidum*, *B. longum* subsp. *longum*, and *B. adolescentis*. Strains representing *Bifidobacterium pseudocatenulatum*, *Bifidobacterium dentium*, and *B. longum* subsp. *infantis* were isolated less frequently (Fig. 1d, e).

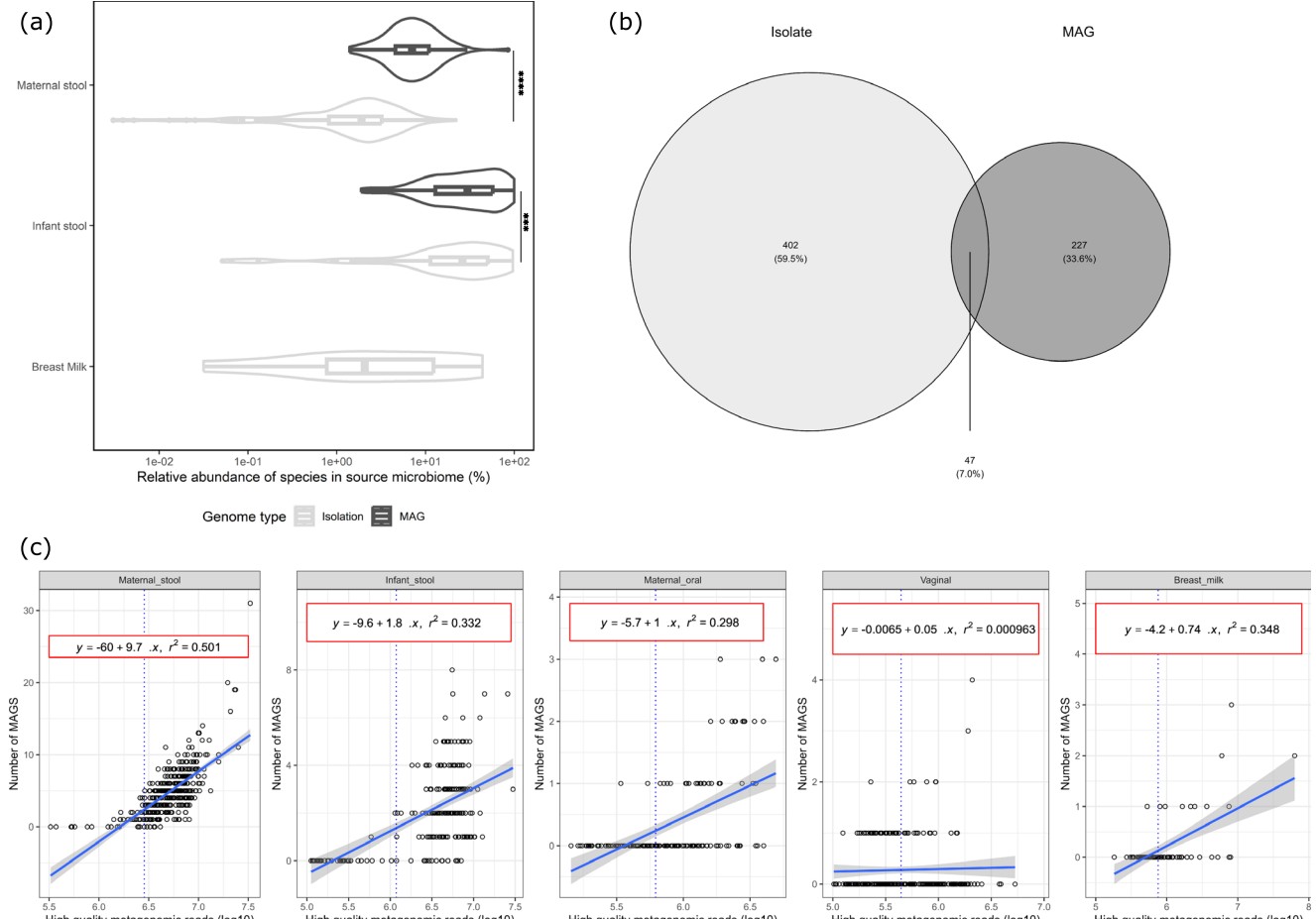

**Fig. 2 | Limit of detection analysis. a** The percentage relative abundance of species for which either metagenomic assembled genomes (MAGs) or culture-isolated genomes were recovered for three sample types. Violin plots visualise the spread of data per sample extending to both the minimum and maximum value. Internal boxplots highlight the middle 50% of data with the median relative abundance shown in the central line. The whiskers extend to show the range of values. Significant differences in groups are annotated with asterisk (***$p = 0.00046$, ****$p = 7.1 \times 10^{-11}$) as determined by two-sided $t$-test. The x-axis is in log scale. **b** Venn diagram showing the number of *Bifidobacterium* species identified by either isolation or metagenomic sequencing, with the number of strains common to both method highlighted in the overlap region. **c** Linear regression models displaying the relationship between MAG recovery and sequencing read depth in metagenomic samples. Solid blue lines indicate the smoothed linear model line with shading visualising the confidence interval around the fit. Individual plots are grouped by sample type and a blue dotted vertical line indicates the minimum read depth required for each sample type in order to recover at least 1 MAG.

Rarely isolated species included *Bifidobacterium lactis* subsp. *animalis*, *Bifidobacterium catenulatum*, *Bifidobacterium angulatum*, and *Bifidobacterium scardovii* (Fig. 1d).

**Selective cultivation of *Bifidobacterium* improves recovery of low abundant species**

Corresponding metagenomic profiling data was available for 388 of the strains isolated via cultivation and was analysed to determine if only strains from the most highly abundant *Bifidobacterium* species in the sample were isolated. The majority (49.7%) of strains isolated via cultivation did not correspond to the most abundant bifidobacterial species in the sample, while 40.2% (156/388) of isolated strains did represent the most abundant *Bifidobacterium* species in the source sample. The remaining 10.1% (39/388) of isolates were not detectable within the metagenomic data.

Next, to further investigate the extent to which metagenomic and culture-based outputs overlapped, we recovered metagenome assembled genomes (MAGs) from the metagenomic sequence data. A total of 274 high quality *Bifidobacterium* genomes were recovered from 218 samples (representing 165 individuals). Half of these were recovered from infant samples

(Supplementary Fig. S1e). Based on the relative abundance of species associated with MAGs and isolated genomes, it appeared that cultivation-based isolation is more sensitive at identifying bifidobacterial genomes in maternal and infant stool samples compared to metagenomic sequencing at the sequencing depth employed (Fig. 2a). In general, a species needed to be present at more than 1% relative abundance to be assembled into a MAG, whereas targeted cultivation was able to identify a species considered to be present at a much lower relative abundance or indeed being undetectable by MetaPhlAn3. In addition, the relative diversity of a sample influenced the ability to recover a MAG of a given species. For maternal stool samples, at least 1.4 million high quality reads were required to recover a MAG, compared to just 588 thousand in an infant stool sample (Fig. 2c). A total of 47/274 *Bifidobacterium* MAGs were found to have an average nucleotide identity (ANI) > 99.9% and genome coverage >90% to the 449 *Bifidobacterium* isolates (Fig. 2b). Taken together, the rather modest overlap between MAGs and cultivation-retrieved genomes suggests that the combined use of methods provides a more complete view of the *Bifidobacterium* strains present in samples.

## Strain micro-diversity is common amongst *Bifidobacterium*

For 80 individuals, multiple strains of a bifidobacterial species were isolated (Supplementary Data 5). We combined these genomes and their equivalent MAGs to elucidate the strain diversity within an individual. ANI analysis revealed that most individuals carry strains that are highly similar, i.e., ANI > 99.9%, (Fig. 3a). This suggests that these isolates belong to the same strain lineage. In addition, alignment of core SNP regions confirmed that most individuals harboured near identical strains. The mean SNP distance between highly similar strains (ANI > 99.9%) from a single individual was 20.80 (range: 1–121), revealing that individuals appear to harbour a distinct strain diversity previously described as "micro-diversity"[24] (Fig. 3b). Some samples, particularly maternal stool, were shown to contain two or more distinct bifidobacterial strains of the same species with an ANI less than 99.0%, suggesting that at least two distinctly different strain lineages existed in these individuals. For example, the 34-week stool sample from PB066 was shown to contain three different lineages of highly similar *B. bifidum* strains (Fig. 3c).

## Commonly isolated bifidobacterial lineages

We found that *B. breve* genomes fell into two monophyletic groups (designated group I and group II; Fig. 3d). One of the identified groups (group I) was shown to represent strains from mothers who had been taking strain *B. breve* 702258 as a supplement during pregnancy and up until three-months post-partum, and therefore this was not considered to be a naturally occurring group. A large *B. breve* group, designated group II, encompassed 72 of the 115 *B. breve* strains isolated from 37 dyads. Strains in group II had a mean pairwise ANI of 99.96% and a mean SNP distance of 91.44 compared to mean ANI of 98.70% and mean SNP distance of 7828 for all strains outside of this group, indicating that members of group II are highly similar. Within this group, various subgroups can be identified, frequently containing strains shared between mother and infant. The strains in group II were compared to 93 publicly available *B. breve* genomes (Supplementary Fig. S4b), revealing that five publicly available genome sequences corresponded to group II; four (two infant isolates and two vaginal isolates) from North American studies and one infant isolate from an Irish study, none of which were previously identified as transmitted from mother to infant or are known probiotic strains.

Long read genome sequencing was carried out on eight *B. breve* strains from group II (selected to represent the sample diversity in the group and to include two transmitted strain pairs) to facilitate full genome assembly to allow for detailed analysis of their homogeneity/the exact genomic differences among members of this group. Alignment of these 8 fully sequenced (i.e., assembled into a single contig) genomes from this study, plus the previously isolated strain 139W423 (Fig. 3e) also found to belong to group II, indicates that the strains are extremely similar with the main differences being: an inversion of an 88 kb genomic region downstream of a predicted *recR* gene in all strains when compared to 139W423, a putative integrative conjugative element (ICE) present in *B. breve* isolate MB0306, and a predicted prophage region present in three strains – 139W423, MM0267 and MB0306 (Fig. 3e). These data highlight subtle differences between these otherwise (essentially) identical strains.

## Very low abundance maternal *Bifidobacterium* strains are transmitted to infants

Whole genome comparisons and core genome SNP-based phylogenetic reconstructions were employed to assess if a strain had transferred from mother to infant. Twenty-seven strain pairs, each representing two isolates from 24 mother-infant dyads, were shown to exhibit an ANI of >99.9 % and were thus considered as transmitted from mother to infant (Fig. 4a). The average SNP difference between mother-infant strain pairs was 22.79 (range 3–61; Fig. 4b) and all strain

pairs appeared on the same subclade of a phylogenetic tree and therefore are presumed to belong to the same strain lineage (Supplementary Fig. S2a). Moreover, SNP differences observed between strains were in the same range as those found within a single sample.

The relative abundance of each transmitted species in the sample showed that 11 of these 27 presumed transmitted strains had not been detected by MetaPhlAn3 in either maternal or infant samples at the sequencing depth obtained by shotgun metagenomics.

Metagenomics-based and whole genome-based methods of strain transmission identification were compared. StrainPhlAn3 and inStrain were used to identify transmitted strains using (meta) genomic information. Transmitted strains were identified using inStrain with a popANI cut-off of 99.999% and genome coverage over 50%. Using only metagenomic sequencing data, ten strain pairs from ten dyads were identified as shared between mother and infant samples. These ten strain pairs had an average popANI of 99.9997%. Eight of these transmitted strains were identified between maternal stool and infant stool, while the remaining two were identified across vaginal samples and infant stool samples. Three of the pairs corresponded to pairs also identified by cultivation whereas the remaining six were newly identified shared strains. Where a shared strain was identified by isolation but not by inStrain analysis, mapping of metagenome reads to genomes from isolated strains revealed that both a low average number of mapped reads per base and low genome coverage were reasons for inStrain's inability to identify such sharing events (Fig. 4c). Isolations alone did not always capture the same bifidobacterial species in mother and infant samples, so inStrain analysis was also performed using sequencing reads from whole genome sequences and thus it was possible to identify metagenomic reads from mother or infant samples that matched to genomes of the corresponding dyad partner, resulting in the additional identification of seven transmitted strains (Fig. 4c).

As shown above, identifying transmitted strains by comparing large genomic regions is difficult for low abundance species. Thus, to potentially uncover more strain transmission events, StrainPhlAn3 was applied to the metagenomic sequencing data combined with whole genome data. As StrainPhlAn3 may be overrepresenting some strain transmission events when run with a thresholding of normalised phylogenetic distance (nPD) of 0.01 (as strains identified by isolation had a nPD less than 0.001 (Supplementary Fig. S2c)), we used a nPD cut-off of 0.001. This combined approach identified 31 putatively transmitted strains (Fig. 3d), greater than the ten identified by StrainPhlAn3 alone.

StrainPhlAn3 detected the highest number of *Bifidobacterium* strain transmissions ($n = 31$) followed by isolations ($n = 27$) and inStrain ($n = 10$). There was considerable overlap of transmitted strains detected between StrainPhlAn3, inStrain and isolations (Fig. 4c, d). Strain isolations detected the greatest number of unique *Bifidobacterium* strain transmissions ($n = 16$) followed by StrainPhlAn3 ($n = 10$), while inStrain only detected one unique transmitted strain. Bacterial isolations recovered strains from maternal stool samples that were not detected by any other method highlighting the sensitivity of this approach. Using a combination of both inStrain and strain isolations, five transfer events were identified that could not be identified by one of these methods alone.

Ultimately, by combining all methods described above, 53 bifidobacterial strains, present in 46 dyads, were identified as transmitted from mother to infant, representing transmission in 34% of dyads (Fig. 4c, d).

## *Bacteroides* and *Bifidobacterium* are the genera most frequently transmitted from mother to infant

Having determined an appropriate threshold for identifying a *Bifidobacterium* strain transfer event, we proceeded to characterise the sharing events between related mother and infant dyads for all species

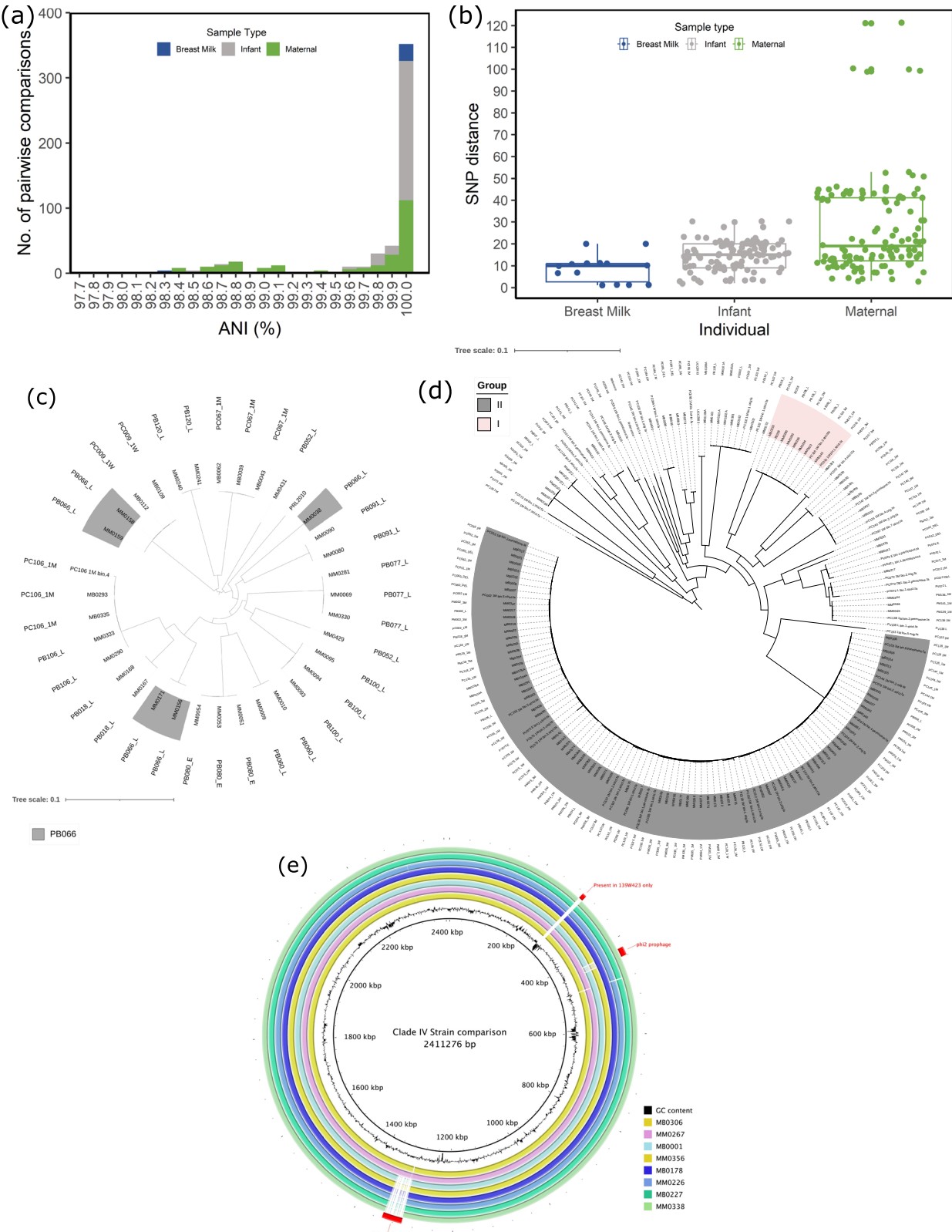

**Fig. 3 | Intrasample strain diversity. a** Pairwise average nucleotide identity (ANI) between strains of the same species in a single sample of maternal stool, infant stool, and maternal breast milk. **b** Pairwise SNP distance between strains of the same species in an individual sample of maternal stool ($n=122$), infant stool ($n=108$), or maternal breast milk ($n=14$) with an ANI ≥ 99.9%. Boxplots visualise the middle 50% of data with the median relative abundance shown in the central line. The whiskers extend to show the range of values excluding the outliers. **c** *B. bifidum* from individuals with multiple strains. Strains from PB066 34-week stool sample are highlighted. **d** Midpoint rooted phylogenetic tree of *B. breve* strains from this study. **e** Comparison of 8 complete *B. breve* genomes from group II.

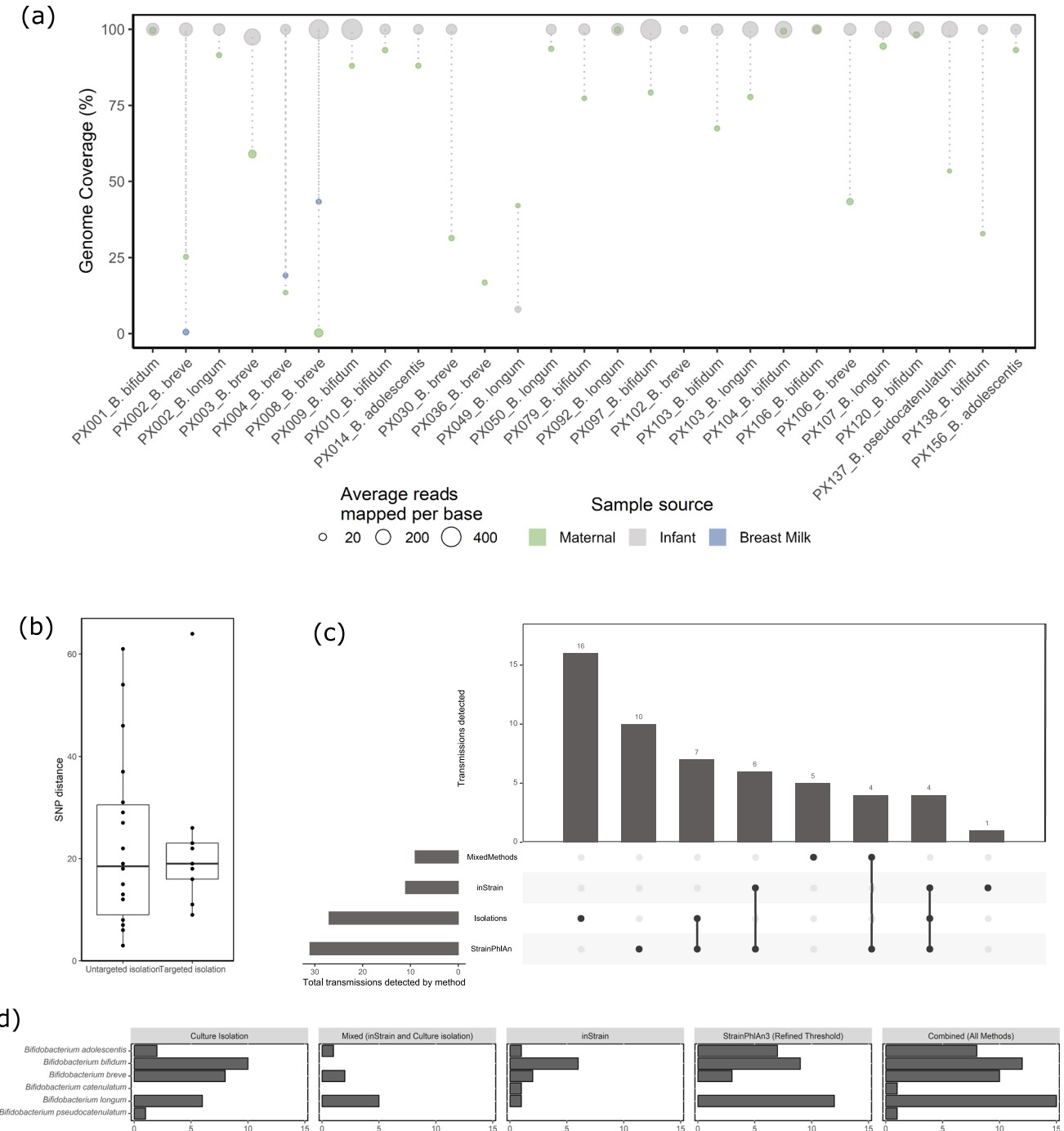

**Fig. 4 | Evidence for strain sharing between dyads. a** Genome coverage of metagenome data mapped to isolated genomes. Size of circle is relative to average number of mapped reads per genome. **b** SNP distance between maternal strain and corresponding infant strains isolated by either targeted ($n = 9$) or untargeted ($n = 18$) approaches. Boxplots highlight the middle 50% of data with the median relative abundance shown in the central line. The whiskers extend to show the range of values excluding outliers. **c** Upset plot showing how many strain transmission events were identified by each method and their overlap. Horizontal bars show the total number of transmission events detected by each method. Vertical bars show the number of transmission events detected by either one or more methods, as represented by multiple dots connected by a filled line. **d** Number of transmitted *Bifidobacterium* strains identified by each method.

with the same cut-off (Supplementary Fig. S4a). The approach combined the isolated genome sequencing with inStrain and StrainPhlAn3 analysis of metagenomics for both *Bifidobacterium* and non-*Bifidobacterium*. In total 135 transmission events were detected across all dyads representing evidence of mother-to-infant bacterial transfer in nearly half ($n = 66/135$) of the studied dyads (Figs. 5a and Supplementary Fig. S3a). This analysis revealed that after *Bifidobacterium*, the *Bacteroides* genus was the second most shared between dyads. For this

genus, there was evidence of strain transfer involving 13 distinct species, including *Bacteroides vulgatus*, *Bacteroides dorei*, and *Bacteroides uniformis* in particular (Fig. 5a). In terms of transmission source site, stool was the most common source of sharing from maternal samples. There were six transmission events observed from the maternal vaginal samples to infant stool and a single strain of *Veillonella parvula* was common to both a maternal oral sample and paired infant sample (Supplementary Data 3).

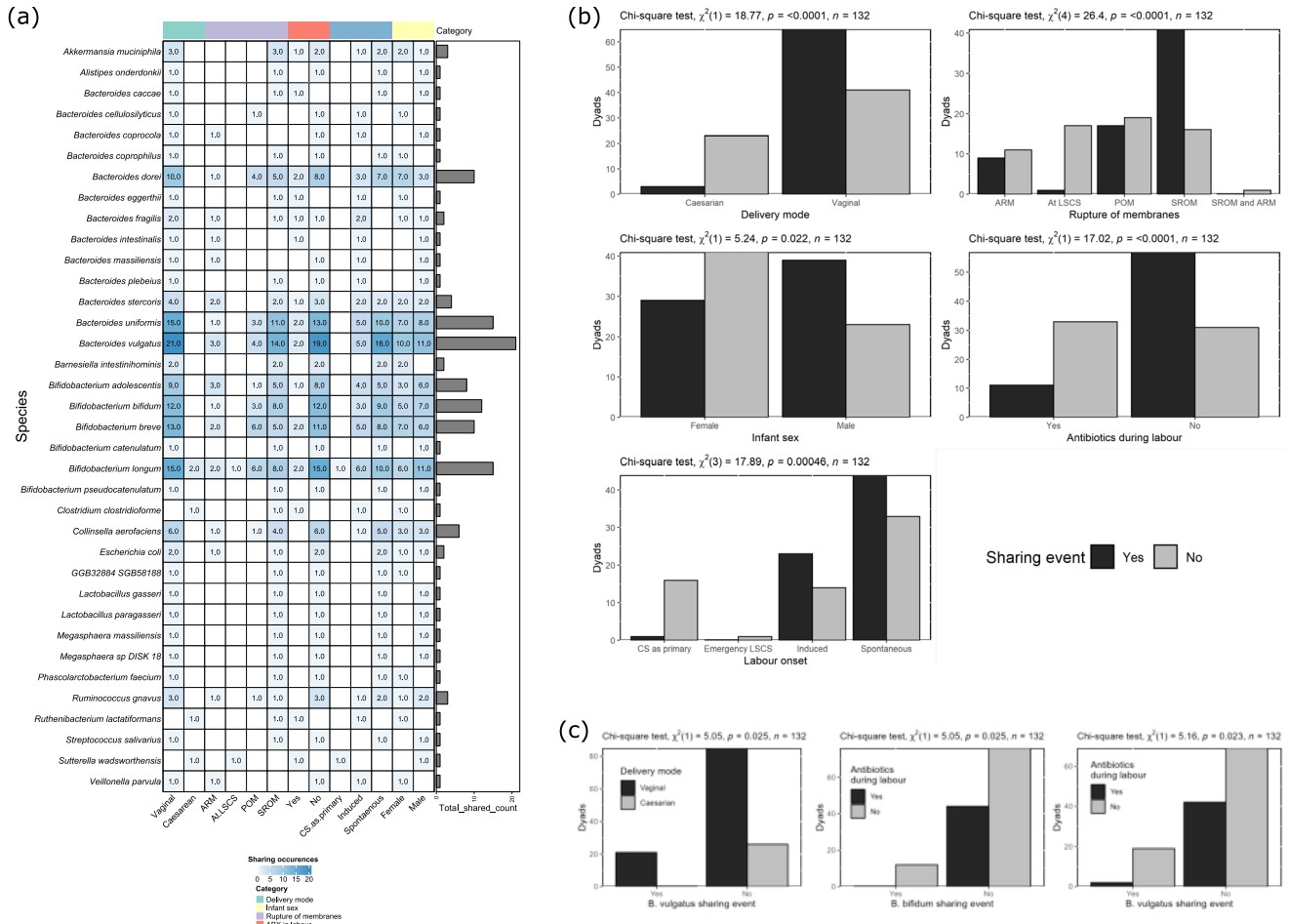

**Fig. 5 | Strains transferred and the factors affecting this transfer. a** Heatmap summary of all species found to transfer from mother to infant and their distribution within each of 5 covariate categories. The total number of sharing events for a given species is represented by the bar plot to the right of the heatmap. Infant male/female split is 62/70, respectively. **b** Each barplot panel represents a different covariate that was significantly associated with sharing of microbes between related mothers and infants. Significant differences between groups were determined using two-sided, Chi-square test with no correction. Infant male/female split is 62/70, respectively. **c** Each barplot panel represents the species whose strain transfer was significantly associated with the given covariate. Significant differences between groups were determined using two-sided, Chi-square test with no correction.

With respect to infant age, the likelihood of detecting a maternal strain was higher in the infant one-month samples. A total of 56/118 (47.5%) samples from infant stools at one-month had a matched strain with any of its maternal samples, compared to just 2/53 (2.77%) infant delivery samples (Supplementary Data 3). The sharing of strains between mother and infant samples collected post-partum was also examined. Using all detection methods, 44 strains from 34 dyads were shared between mother and infant after birth. Like the transferred strains, members of the genera *Bifidobacterium* and *Bacteroides* were shown to be most frequently shared (Fig. 5a and Supplementary Fig. S3d). Over half (31/44) of these strains were shared between maternal stool and infant stool, and the remaining 13 strains were shared between breast milk and infant stool (See Supplementary Data 4).

Finally, we investigated if the transmission of any of the 135 strains from the mother had an influence on the overall infant microbiome at one-month. From the 36 transferred species, *B. bifidum, B. breve,* and *E. coli* had a significant association with the beta diversity of the infant stool microbiome at this timepoint as determined using envfit (Supplementary Fig. S3b). Whilst this effect appeared independent of the relative abundance (Supplementary Fig. S3c), no distinct signature change in the microbiome of these samples was apparent (Supplementary Fig. S3d).

## Delivery mode is a key factor in strain transmission

To determine the impact of various maternal, infant, or other environmental factors on transmission of strains from mother to infant, Fisher's tests were performed on 22 different covariates with the strain sharing events detected by our combined genome sequencing and inStrain approach. From these, just five were significantly associated with strain sharing (Fig. 5b). Delivery mode was found to be very strongly associated with strain sharing events (Chi-square test $p \leq 0.001$), with dyads from vaginal births accounting for 95.45% of events. The diversity of species shared from vaginal births was also greater than caesarean births, with just seven different shared species observed in caesarean births compared to 26 for vaginal births. Three of these, *Sutterella wadsworthensis, Ruthenibacterium lactatiformans* and *Bifidobacterium longum* subsp. *infantis* were not shared in vaginal births (Fig. 5a). The influence of delivery mode on strain sharing was also reflected in a significant association with membrane rupture. Strain sharing occurred most frequently in instances where there was a spontaneous rupture of membranes (SROM; $n = 41$ dyads) and was most infrequent in instances of caesarean birth ($n = 1$). Puncture of membranes (POM; combined term and preterm) had a negative effect on the number of strain sharing events (Fig. 5b). In concordance with this, the type of labour onset was also significantly associated (Chi-square test $p \leq 0.001$) with strain sharing, whereby spontaneous labour

had the greatest number of sharing events ($n = 44$), followed by induced labour ($n = 23$). Finally, maternal exposure to antibiotics during labour had a significantly negative influence on the number of shared events, with fewer strains shared. In particular, the profile of shared species was different, with only a small number of *Bacteroides* and *Bifidobacterium* being identified as shared from mothers administered antibiotics during delivery. Specifically, sharing of *Ba. vulgatus* was significantly greater in vaginal delivery compared with instances of caesarean section and found to occur less frequently when the mother was administered antibiotics in labour (Fig. 5c). In addition, *B. bifidum* sharing was significantly higher in the absence of antibiotics in labour (Fig. 5c).

## Discussion

The ability to accurately identify microbial strain transfer from mothers to infants is dependent on the ability to recover enough sequence data from either source. Shotgun sequencing has been used to investigate this phenomenon with generally good sequence coverage obtained; however, this approach is limited to bacteria that are present at a relatively high abundance[7,8,25]. Studies using cultivation-based methods have provided better resolution than initial shotgun-based approaches and yet this approach is limited to culturable species and is laborious[16,26]. Cultivation-based methods reveal the strain micro-diversity within an individual[24,27] and we have shown here that this is greater in adults than infants. This is likely due to slow evolution of strains in adults[27]. We considered this micro-diversity and included phylogenetic analysis rather than strict nucleotide identity thresholds alone to identify mother and infant strains that belonged to the same lineage and thus highly likely to have transferred from mother to infant.

We combined the genomes of isolated strains with metagenomic data to provide a high resolution, validated, image of bifidobacterial strain transmission. This approach identified strain sharing in numerous dyads in the study population and, importantly, proved an invaluable method to detect lower abundance species/strains. Indeed, we identified 16 strain sharing events that would have been overlooked if a complementary culture-based approach had not been used, i.e., the strains were cultured from stool but not identified through metagenomic sequencing. Moreover, culture-based was essential for identifying bifidobacterial strains in breast milk. To improve detection of transmitted strains, we employed a targeted approach to strain isolation[28] whereby genetic and phenotypic information about the infant bifidobacterial strain allowed the isolation of that strain from the corresponding maternal sample. This was possible due to extensive prior knowledge about *Bifidobacterium* and the approach could be extended to other taxonomic groups as knowledge of cultivation methods improves. Furthermore, the use of robotics to automate some of the process could significantly improve throughput[29].

The use of two complementary bioinformatics methods to detect strain transmission, StrainPhlAn3 and inStrain, while revealing significant overlap, allowed the identification of additional putative strain transmissions. Specifically, metagenomic methods identified 23 transmitted strains that were not shown by cultivation alone due to the low throughput of strain isolations. Additionally, five transmitted strains were specifically (and exclusively) identified by combining results from inStrain and culture isolations. The use of both culture-based and metagenomic approaches provided a more detailed picture of strain transmission than previously shown. This is apparent in the *B. breve* strains that were only shown to be transmitted by cultivation methods and would otherwise have been overlooked due to their low abundance in maternal stool. Thus, use of both methods together is optimal and, when augmented with the evidence of strain transfer from direct culture work, provides a more detailed picture of strain transmission.

Using this verified combined approach, we investigated strain transmission patterns for all members of the microbiota. In agreement with previous studies, strains of *Bacteroides* and *Bifidobacterium* were most frequently shared between mothers and infants[7,25,30]. Interestingly, we found that strains of *Bifidobacterium* can be transferred at a relatively high frequency despite a clear difference between the bifidobacterial profile of mothers and infants. The difference in species dominance seems to be primarily driven by dietary carbohydrate differences[31,32]. Regardless of this difference in dominance, it is clear that many of the infant-associated species are present in the mother at an abundance that is too low to facilitate detection by shotgun metagenomics alone, at the sequencing depth employed. In most cases, this frequency of *Bifidobacterium* transfer would not have been observed without the targeted culture approach.

Like previous metagenomic-based studies[8,13,26], we have identified that most transmitted strains are initially found in maternal stool, but that some were also present in breast milk. A small number of strains shared between infant stool and breast milk were not identified in maternal stool, despite attempts to isolate these strains. Thus, it is not clear if these infants inherited these strains from their mother, both mother and infant have acquired them from secondary sources, or the mother has acquired them from the infant.

In summary, by combining metagenomic sequencing with culture and whole genome sequencing, our study has provided the most in-depth investigation to date of the transmission of *Bifidobacterium* from mother to infant in early life. Culturing of these strains confirms their viability in the host from which they are isolated and makes these strains available for further assessment and application. The resolution we obtained shows that maternal-to-infant transmission is a common phenomenon and is strongly influenced by external factors including mode of delivery. While it is known that maternal microbiome can influence rates of prematurity[33], the maternal gut microbiome is also key to the establishment of neonatal gut health. The value of culture work to complement sequencing is very high and should be strongly considered for any future research in this area.

## Methods

### Participant recruitment and sample collection

Participants were recruited at the National Maternity Hospital, Dublin, Ireland as part of the MicrobeMom study between 15th September 2016 and 12th July 2019 (ISRCTN53023014)[20]. Ethical approval for the study was received from National Maternity Hospital research ethics committee in February 2016 (EC 35.2015). Written informed consent was obtained from all participants and the study was completed in accordance with the Declaration of Helsinki. Where possible, stool, vaginal, and oral rinse samples were collected from mothers at 16- and 34-weeks or pregnancy. In addition, stool samples were collected from mothers at one-month and three-months post-partum. An expressed breast milk sample was also collected from mothers at one-month. Infant stool samples were collected at delivery, within one-week, one-month, and at three-months post-partum. Numbers and description of samples collected and analysed are reported in Fig. 1a and Supplementary Data 1. A sub-sample of stool (7 g) was added to RNA*later* and along with the remaining sample was stored at −80 °C. As an environmental control, RNA*later* was added to stool-free sample pots and frozen. Breast milk samples were collected by pump expression, aliquoted, and stored at −80 °C. Vaginal swabs were collected from the posterior fornix using dry cotton swabs and stored at −80 °C. For oral samples, mothers swirled 5 ml of sterile PBS in their mouths and expectorated into sterile tubes. Following centrifugation at $10,000 \times g$ for 10 min, the supernatant was removed, and samples were frozen at −80 °C.

### Nucleic acid extraction

DNA was extracted from all the RNA*later* fixed stool samples within eight-weeks of sample collection using AllPrep DNA/RNA kit (Qiagen). Briefly, 100 mg of faecal sample was centrifuged at maximum speed

for 10 min and excess RNAlater and supernatant was removed. The pellet was resuspended with 100 µl bacterial lysis buffer (30 mM Tris-HCl, pH 8.0, 1 mM EDTA plus 15 mg/ml lysozyme) and 10 µl proteinase K (20 mg/ml)). Following incubation, 1.2 ml Qiagen RLT Plus buffer containing 1% β-mercaptoethanol was added to the sample and vortexed briefly. Samples were transferred to 2 ml sterile bead beating tubes filled with 1 ml of 0.1mm glass beads and subjected to bead beating for 3 min. 700 µl of lysate was added to a QIAshredder spin column (Qiagen) and centrifuged at maximum speed for 2 min. DNA was extracted from the lysed sample using the AllPrep kit (Qiagen) as per manufacturer's instructions. Negative control extraction blanks were included for every 50 samples.

The DNA from oral rinse samples was extracted using the PowerFood kit, whereby sample pellets were resuspended in 1 ml sterile PBS, centrifuged at $13,000 \times g$ for 1 min and resuspended in 450 ml MBL before bead beating for 3 min and subsequently processed according to the manufacturer's protocol. For vaginal swab samples, 1 ml of sterile PBS was added to the swab sleeve and the sample was vortexed vigorously for 2 min. Extraction was carried out on 700 ml of suspension with the remainder frozen in 10% glycerol for future potential bacterial culture. The sample was centrifuged at $13,000 \times g$ for 1 min after which the supernatant was removed. The pellet was resuspended in 450 ml MBL and heated at 65 °C for 10 min with bump vortexing every 2 min, after which sample processing continued according to the manufacturer's instructions for the PowerFood kit.

DNA from breast milk samples was extracted as previously described[34]. Briefly, milk samples were centrifuged at $4500 \times g$ for 20 min at 4 °C. After centrifugation, cream and supernatant were discarded, and resulting pellets were washed twice. Each pellet was resuspended in sterile PBS, centrifuged at $13,000 \times g$ for 1 min, and the supernatant was discarded. After the washing steps, the pellets were resuspended in sterile PBS. DNA was extracted using the MolYsis complete5 kit in accordance with the manufacturer's instructions. The host cells were lysed by the addition of a chaotropic buffer, and the nucleic acids that were released were degraded by an enzyme, MolDNase. The microbial cells were then sedimented and lysed using reagents and proteinase K. Subsequently, microbial DNA was isolated and extracted using spin columns, and 100 µl of DNA was eluted and stored at −20 °C.

## Metagenomic sequencing and quality control

DNA was quantified using the Qubit™ dsDNA HS Assay Kit and normalised to 0.2 ng/ml. Sequencing libraries were prepared according to the Nextera XT DNA Library Preparation Kit (Illumina) protocol and pooled to 2 mM. All libraries were sequenced employing the $2 \times 300$ bp paired end kit executed on an Illumina NextSeq platform. The generated raw data sets were subjected to base calling using Illumina's bcl2fastq software (v 2.19). Adapter removal and quality trimming was performed using TrimGalore (v 0.6.0), a wrapper script for Cutadapt[35] (v 2.6) and FastQC (v 0.11.8), using default parameters. Resulting reads were aligned to the human genome with Bowtie2[36] (v 2.3.4), aligned reads were removed with samtools[37] (v 1.9), and converted from BAM to fastq format with bedtools[38] (v 2.27.1). Interleaved fastq files were generated using BBMap (v 38.22). Samples that did not have any taxonomic assignment were removed from analysis. In addition, any samples with fewer than 100,000 post-filtering reads were removed as these fell below the maximum number of reads observed in negative control samples. Finally, singleton samples (i.e., maternal samples with no corresponding infant sample) were removed.

## Metagenomic data analysis

Compositional and functional profiling was carried out with HUMAnN3 (v 3.0) using ChocoPhlAn nucleotide database (v 30), uniref90 protein database, and MetaPhlAn3 database version (v 30). Samples with no

classified reads were removed from the dataset. In addition, samples that had fewer than 100,000 reads were removed as these fell below the maximum number of reads observed in negative control samples.

Community-level microbiome analysis was performed using the vegan package in R with the MetaPhlAn3 species table as input. Alpha (within-sample) diversity was calculated using Shannon's index and compared between body sites and timepoints by ANOVA. Beta (between-sample) diversity was calculated using Bray-Curtis dissimilarity and compared between body sites and timepoints by PERMANOVA using vegan's adonis function. Clustering of samples was visualised by non-metric multidimensional scaling (NMDS) using vegan's metaMDS function and the relationship between microbiome composition and sample metadata (age, parity, antibiotic exposure etc.) was investigated using vegan's envfit function[22] which performs multiple regression of the metadata covariates with the NMDS ordination axes and generates a $p$ value by permutation. This association was also performed using the relative abundance of frequently transmitted species as covariates to determine whether these transmission events impacted the microbiome composition. Metagenomes were assembled with metaSPAdes (v 3.14)[39] and assembly statistics were calculated using the assembly_stats python library (v 0.1.4; DOI: 10.5281/zenodo.3968774.). To recover metagenome-assembled genomes (MAGs), paired fastq reads were aligned to assembled metagenomes by Bowtie2 with default parameters. The resulting SAM files were converted to BAM format and sorted using Samtools, and contig depth was calculated using the jgi_summarise_bam_contig_depths script bundled with MetaBat2 (v 2.12.1)[40]. MetaBat2 was also used for contig binning and the lineage_wf workflow of CheckM (v 1.0.18)[41] was used to evaluate the quality of metagenomic bins. Only those bins with completeness ≥90% and contamination <5% were considered "high quality" and retained for downstream analyses.

## Assignment of Lewis status and secretor status

A 100 µl volume of breast milk was diluted with 100 µl of water and the lipid was removed via centrifugation at $4000 \times g$ at 4 °C. The aqueous layer was recovered and filtered through a 1 µm glass fibre plate. Proteins were ethanol precipitated and centrifuged, the upper liquid fraction was collected and dried. All samples were reconstituted in 100 µl of water and subjected to a sequential solid phase C18 and Carbograph microplate extraction protocol. The HMO eluent was labelled with 2-aminobenzamide by reductive amination[42] and free label and salts were removed using Diol plates. The separation of 2AB-derivatized HMOs was carried out by UPLC with fluorescence detection on a Waters ACQUITY UPLC H-Class. The HILIC separation was performed using a Waters Ethylene Bridged Hybrid (BEH) Glycan column. The separation was performed using a linear gradient of 88–43% MeCN at 0.56 mL/min over 35 min. An injection volume of 20 µl prepared in 88% v/v MeCN was used throughout. The system was calibrated using an external standard of 2-AB-labelled glucose oligomers to create a dextran ladder as previously described[43]. After lipid extraction, all samples were processed on a 96 well plate using an 8 channel multi pipette. Each sample was processed across three plates. The protocol was validated using commercially available HMO standards and the observed Glucose Unit (GU) values obtained were comparable to those referenced in the literature. 50 reproducible glycan peaks were resolved in all samples and integrated. The constituent HMO peaks were assigned. Secretor status was determined by the presence or near absence of 2'FL and LNFPI and Lewis status was assigned on the relative abundances of 3FL, LDFT, LNFPII, LNFP III and LDFH (Supplementary Data 6).

## Selective isolation of *Bifidobacterium* spp

Stool aliquots were retrieved from storage and allowed to thaw at room temperature before resuspension of in phosphate buffered saline (PBS; Sigma Aldrich) + 0.06% cysteine-HCL (Sigma Aldrich) at a

concentration of 0.1 mg/ml. Samples were thoroughly resuspended by vortexing, then serially diluted (1:10) to $10^{-7}$ and then plated on to the above media supplemented with either lactose (Sigma Aldrich), LNnT or 2′FL (gift from Glycom A/S). Following an initial 48 h anaerobic incubation at 37 °C, isolates were re-streaked to purity. Pure colonies were inoculated into mMRS broth +0.5% (w/v) lactose and incubated overnight at 37 °C in an anaerobic chamber. A biochemical assay for to fructose-6-phosphate phosphoketolase F6PPK activity was performed to indicate presence of *Bifidobacterium* spp., as previously described[44]. Briefly, 0.5 ml of overnight bacterial culture was centrifuged at $4000 \times g$ for 4 min and washed twice in 0.5 ml of phosphate buffer ($0.05 M$ $KH_2PO_4$ supplemented with 0.05% w/v cysteine hydrochloride, pH 6.5). Cells were pelleted as before and resuspended in 100 µl of phosphate buffer plus 0. 0.25% v/v Triton X-100 and incubated at room temperature to allow for cell lysis. The following was then added to lysed cells: 25 µl of fructose-6-phosphate (80 mg/ml) and 25 µl of a solution containing 6 mg/ml NaF and 10 mg/ml Na-iodoacetate. After 60 min incubation at 37 °C, the reaction was stopped with 150 µl of 13.9% (w/v) hydroxylamine-HCl, freshly neutralised with NaOH to give a pH of 6.5. After 10 min at room temperature, 100 µl of 15% (w/v) trichloroacetic acid, and 100 µl of 4 M HCl were added. Colour development was achieved by adding 100 µl of 50 mg/ml $FeCl_3.6H_2O$ in 0.1 M HCl. Immediate colour change to red-violet colour develops immediately this is taken as a positive indicator for the presence of F6PPK.

Bifidobacterial strains were routinely cultured in mMRS +0.5% lactose, 100 µg/ml mupirocin and 0.05% cysteine-HCl; for solid media, agar was added at 1.5% (w/v).

### Targeted isolation of *Bifidobacterium* based on carbohydrate utilisation

Where a strain had been isolated from an infant or breast milk sample but not a corresponding maternal stool sample, the genomic data from the infant strains and the knowledge that *Bifidobacterium* species can utilise a wide array of carbohydrates was exploited to attempt isolation of the same strain from maternal stool, similar to a method described previously[28]. Strains were isolated as previously described but the carbohydrates used were ribose (for *B. breve*; Carbosynth), arabinose (for *B. longum* subsp. *longum* isolations; Carbosynth) and 2′FL (for *B. bifidum* and *B. longum* subsp. *infantis* isolations; a gift from Glycom A/S). The presence of CRISPR-Cas protospacer sequences in the genomes of infant strain was predicted using crisprViz. Where protospacer sequences were predicted, strain-specific primers were designed based on unique protospacer sequences using primer3plus. Where no CRISPR-Cas regions were predicted, a species-specific primer was used. See Supplementary Table S6 for list of primers used.

### Bacterial genomic DNA extraction

DNA was extracted from 1.5 ml of overnight bacterial culture grown in mMRS +0.5% (w/v) lactose using GenElute bacterial DNA isolation kit (Sigma-Aldrich) according to manufacturer's instructions. Briefly, cells were centrifuged at $4000 \times g$ for 3 mins and resuspended in 200 µl lysozyme solution (Gram positive lysis solution supplemented with $2.115 \times 10^6$ unit/ml lysozyme from chicken egg white (Sigma-Aldrich)) and incubated for 30 min at 37 °C. 20 µl RNaseA was added and incubated for 2 min at room temperature. 20 µl of proteinase K (20 mg/ml) plus 200 µl of lysis solution C was then added and cells incubated for 10 min at 55 °C. 200 µl of 96% EtOH was added to cells and resulting solution loaded into a GenElute Nucleic Acid Binding Columns. Following binding and washing, DNA was eluted into 120 µl of Elution solution (10 mM Tris-HCl, 0.5 mM EDTA, pH 9.0).

DNA isolations for Pacific Bio (PacBio, Menlo Park, CA, USA) sequencing was performed using NucleoBond AXG 100 columns and

NucleoBond Buffer Set III (Macherey-Nagel, Düren, Germany) with minor modifications to the manufacturer's instructions. A single colony was inoculated into mMRS +0.5% lactose, 100 µg/ml mupirocin and 0.05% cysteine-HCl and incubated overnight at 37 °C in an anaerobic chamber. Cells were then diluted 1:50 into the same media and grown anaerobically at 37 °C until mid-log phase (0.5–0.8 $OD_{600nm}$) and pelleted by centrifugation at $5000 \times g$ for 10 min. Pellets were frozen at −80 °C and re-suspended in 5 ml Buffer G3 plus 20 mg/ml lysozyme (Sigma-Aldrich) and 250 U mutanolysin (Sigma-Aldrich). Cells were incubated for 1 h at 37 °C following which 1.2 ml of Buffer G4 was added and incubated for 1 h at 55 °C. Cell debris was removed through centrifugation at $5000 \times g$ for 10 min and the clear lysate transferred to the NucleoBond AXG 100 column. Manufacturer's instructions were then followed, and DNA resuspended in 10 mM Tris Buffer (pH 8.0).

### Genomic DNA sequencing, assembly, and annotation

Short read DNA sequencing was performed by GenProbio s.r.l. (Parma, Italy) using an Illumina HiSeq platform. Sequencing reads were assembled using Spades (v 3.1). Long reads were sequenced using either PacBio RS II platform (strains: MM0267, MB0001, MB0306, MM0356, MB0178, MM0226, MB0227) or Pacbio Sequel II (strain MM0338). A hybrid assembly of Pacbio and Illumina sequencing reads using Unicycler (v 0.4.8) was performed for strains sequenced with Pacbio RSII platform. Strains sequenced by Pacbio Sequel II platform were assembled using canu (v 2.2.1) and HiFi reads only. Complete chromosomes were manually trimmed and rotated using circulator (v 1.5.5).

Open Reading Frame (ORF) prediction and automatic annotation was performed using Prodigal (v 2.6.3) (http://prodigal.ornl.gov) for gene predictions, BLASTP (v 2.6.0) (cut-off E-value of 0.0001) for sequence alignments against a combined bifidobacterial genome-based database, and MySQL relational database to assign annotations. Predicted functional assignments were manually revised and edited using similarity searches against the non- redundant protein database curated by the National Centre for Biotechnology Information (ftp://ftp.ncbi.nih.gov/blast/db/) and PFAM database (http://pfam.sanger.ac.uk), which allowed a more detailed, in silico characterisation of hypothetical proteins. GenBank editing and manual inspection was performed using Artemis (v 18) (http://www.sanger.ac.uk/resources/soft-ware/artemis/). Transfer RNA genes were identified employing tRNAscan-SE (v 1.3.1) and ribosomal RNA genes were detected based on the software package Rnammer (v 1.2) supported by BLASTN (v 2.6.0).

Whether the isolated strain was the dominant or non-dominant bifidobacterial species in that sample was calculated as follows: For each isolated genome, the relative abundance of that species was obtained from the results table of MetaPhlAn3 analysis and compared to the relative abundance of all *Bifidobacterium* species in that sample. If an isolated strain was from the species with the highest relative abundance, it was classified as the dominant species and if not, it was classified as the non-dominant species. If the relative abundance for all bifidobacteria was zero, then it was classified as not detected.

### Strain classification and SNP region phylogenetic reconstructions

Whole genomes from isolated strains were used to classify strains into respective species. Whole genomes were compared to a local database of genomes of 66 *Bifidobacterium* species using fastANI (v 1.3)[45] with default settings. A threshold of 98 % average nucleotide identity was used to classify bifidobacterial species.

SNP differences in the core genome of species was determined as follows. Raw paired end sequencing reads were trimmed with trim-momatic (v 0.36)[46] with the following options "-phred33

ILLUMINACLIP:TruSeq3-PE.fa:2:30:10 LEADING:3 TRAILING:3 SLI-DINGWINDOW:5:20 MINLEN:50". For each strain, paired trimmed reads were aligned to a species-specific reference strain using snippy (v 4.4.5) (https://github.com/tseemann/snippy). The core genome alignment of all strains of a species was generated with snippy-core and cleaned with snippy-clean_full_aln. Gubbins (v 3.0.0)[47] was used to identify and remove putative areas of recombination with the following options "-m 4 -b 4000 --first-tree-builder fasttree". SNP sites were identified with snp-sites (v 2.5.1) and the resulting file used to generate a phylogenetic tree with RAxML (v 8.2.12)[48] as follows: 20 maximum likelihood (ML) trees (parameters: -m ASC_GTRGAMMA --asc-corr=lewis -p 12345 -N 20) and 100 bootstrap trees (parameters: -m ASC_GTRGAMMA --asc-corr=lewis -p 12345 -x 12345 -N 100) were used to generate the best tree (parameters: -m ASC_GTRGAMMA --asc-corr=lewis -p 12345 -f b). Phylogenetic trees were visualised and annotated in iTOL[49].

### Identifying transmitted strains using whole genome sequence data

Average nucleotide identity between all strains was calculated using fastANI (v 1.3). Resultant data were imported into R and combined with metadata on strain source and strain species. The pairwise ANI was plotted using ggplots package in R. Results from fastANI analysis outlined above was interrogated to reveal strains from the same dyad that had an ANI ≥ 99.9%. Then the SNP differences between these strains was identified by aligning the trimmed paired sequencing reads from an infant sample to an assembled draft genome of a maternal sample using snippy (v 4.4.5). Finally, the phylogenetic tree of the core SNP regions of (outlined above) was examined to identify if strains from the same dyad formed monophyletic groups. An infant strain with fewer than 200 SNPs when compared to the corresponding maternal strain and that also formed monophyletic groups was deemed to have transferred from mother to infant.

### Strain-level profiling by StrainPhlAn3

Strain-level profiling using StrainPhlAn3 was performed using default parameters unless specified. Briefly, all MetaPhlAn3 marker genes were recovered from samples, and, for every species detected, marker genes were extracted and compared between samples using the --markers_in_clade threshold of 0.5. Shared strains were identified by using a normalised phylogenetic distance determined as follows: StrainPhlAn3 was run with nucleotide sequence files of genomes of bifidobacteria isolated from maternal stool, breast milk and infant stool along with metagenomic sequences. The resulting phylogenetic trees were normalised by their total branch length and converted to a phylogenetic distance matrix using the *ape* package (v 5.5) in R. The phylogenetic distance for each bifidobacterial strain pair was extracted and plotted using ggplot. All bifidobacterial strains previously identified as identical using ANI and core SNPs had a normalised phylogenetic distance less than 0.001 and therefore this value was used as a threshold for identifying shared strains using StrainPhlAn3.

### Strain-level profiling by inStrain

Strain-level profiling using inStrain was performed using both metagenomic and isolate fastq files as input. The database was constructed from the previously recovered MAGs after dereplication at 98% identity using dRep[15] and alignments were performed using Bowtie2. The genome identity and coverage thresholds recommended by the authors, 99.999 and 50% respectively, were used to identify strain sharing events.

### Statistics and reproducibility

This study is a subsequent analysis of data collected as part of the MicrobeMom trial[20] and no statistical method was used to determine a sample size for this analysis. Investigators were blinded while the original trial was conducted but were unblinded during the analysis of the data described in this manuscript. For metagenomic analysis, samples were excluded that did not have any taxonomic assignment or had fewer than 100,000 post-filtering reads. In addition, singleton samples (i.e., maternal samples with no corresponding infant sample) were removed. Statistical analysis in the manuscript was performed in R using base R and the vegan package.

### Reporting summary

Further information on research design is available in the Nature Portfolio Reporting Summary linked to this article.

## Data availability
All raw sequencing data used in this study is available in the ENA repository under accession number PRJEB48251.

## Code availability
Scripts used in the analysis are available via GitHub (https://github.com/SligoMicrobe/MicrobeMom).

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

## Acknowledgements

This publication has emanated from research conducted with the financial support of Science Foundation Ireland (SFI) under Grant Numbers SFI/12/RC/2273 and 16/SP/3827, an Irish Research Council fellowship (EPSPD/2016/25) with Alimentary Health Ltd awarded to C.F, and from the European Union's Horizon 2020 research and innovation programme under the Marie Sklodowska-Curie grant agreement No. [754535] awarded to I.J.O.N. Glycom A/S are thanked for the kind gift of synthetic LNnT and 2'FL. We would like to thank all the mothers and infants recruited to the MicrobeMom study for their generous time and engagement.

## Author contributions

F.S., F.M.M., D.V.S., P.D.C. conceived the project, C.F., I.J.O.N., C.J.W., R.L.M., S.L.K., A.A.G., E.M., D.B., R.S.-G., S.R.C.N., I.B.N., E.W., E.M., R.O.F. designed and performed the experiments and collected the data. C.F., I.J.O.N., C.J.W., D.V.S., P.D.C. analysed and interpreted the data. P.M.R., D.G., F.S., R.S., F.M.M., D.V.S., P.D.C. supervised the project. C.F., I.J.O.N., C.J.W. wrote the first draft of the paper and performed subsequent revisions. All authors reviewed, edited and approved the final version of the paper.

## Competing interests

D.G. is an employee of PrecisionBiotics Ltd (Novozyme Cork) who sponsored the clinical study. All other authors declare no competing interests.

## Additional information

[1]Teagasc Food Research Centre, Fermoy, Co, Cork, Ireland. [2]APC Microbiome Ireland, National University of Ireland, Cork, Ireland. [3]Nuffield Department of Medicine, University of Oxford, John Radcliffe Hospital, Oxford OX3 9DU, United Kingdom. [4]School of Microbiology, University College Cork, Cork, Ireland. [5]UCD Perinatal Research Centre, School of Medicine, University College Dublin, National Maternity Hospital, Dublin, Ireland. [6]SFI Centre for Research Training in Genomics Data Science, School of Mathematics, Statistics & Applied Mathematics, University of Galway, Galway, Ireland. [7]NIBRT GlycoScience Group, National Institute for Bioprocessing Research and Training, Co, Dublin, Ireland. [8]Department of Chemistry, Maynooth University, Maynooth, Co, Kildare, Ireland. [9]Bioprocessing Technology Institute, AStar, Singapore, Singapore. [10]PrecisionBiotics Group Ltd. (Novozymes Cork), Cork Airport Business Park, Kinsale Road, Cork, Ireland. [11]UCD School of Medicine, College of Health and Agricultural Science (CHAS), University College Dublin (UCD), Dublin, Ireland. [12]These authors contributed equally: Conor Feehily, Ian J. O'Neill, Calum J. Walsh. [13]These authors jointly supervised this work: Fionnuala M. McAuliffe, Douwe Van Sinderen, Paul D. Cotter. ✉e-mail: d.vansinderen@ucc.ie

