## [Peer Review File · Nature Communications]

REVIEWER COMMENTS

Reviewer #1 (Remarks to the Author):

Feehily et al. present an interesting manuscript examining the transmission of Bifidobacterium strains from mother-to-child using metagenomic and culturomic approaches. Their cohort, and the amount of isolated strains are very impressive and are surely in the great interest of the field. However, the story in this manuscript is not clear, and the analysis is not thorough enough, or maybe simply not explained. I hope the authors will take the suggested feedback and produce a revised manuscript that will enable them to highlight their unique cohort and isolates.

Major comments:

- The paper reads like a list of small details, without a clear storyline. The text is jumping from one topic to another without any deep insights in each. A better description of the transferred strains, compared across delivery modes, perhaps even with a functional analysis, will be much more informative, and can teach the community about the factors that impact infant colonization in early life.
- There are many statistically significant p-values in the text, but it is unclear what they stand for, or how they were calculated. For example, "Parity was observed to be significantly associated with the vaginal microbiome ($p=0.0019$)". What does this mean? Is it associated with an increase of specific bacteria? Decrease of others? The reader cannot take away any knowledge from this. There is no mention of a figure showing this comparison, or analysis clearly, so the results section read as a list of dead-end streets.
- The figures are not readable. Not in their font size, nor in their attempt to convey a message. For example, in Figure 1: (a) it is unclear what the "|" mean in the numbers, also, which infants were born vaginally or by CS? (b) showing prevalence and abundance on the same plot is very confusing, two small panels are better than this combined view. (c) What are the different blocks of the phylogenetic tree? It is never explained. In Figure 2F the colors are never explained, they use the same pallet of colors in f and g and as far as I understand, for two different reasons. Figure 3b is again re-using the same two colors only each time for a different variable. All these issues are making the figures really hard to interpret (with and without the accompanying text).
- The mother and infants have different bifidobacterium species, a discussion of this difference in light of transmission will be helpful.

Minor comments:

- Throughout the manuscript the amount of reads per metagenomic sample is not mentioned. This is important since the authors claim that MAGs did not cover a lot of the bacteria present in the sample. Perhaps deeper sequencing would solve that.

- In addition, in lines 149-151, it is mentioned that "372.50-fold genome coverage is a very high coverage". With this coverage, we would expect significantly less than 26 contigs per genome. Are there a lot of outliers? What is the median coverage?

- Figure 2f: This is not a venn diagram as written in the caption. What do the bars represent? Not clear.

- Figure 3b: same colors used for different things

- Line 119 refers to fig1d, but the vaginal microbiome is not described there (to the best of my understanding)

- B. breve: why were these 9 strains sequenced, they seem to be the same on the phylogenetic tree? Why were any differences expected to be found? Once none were found why is this interesting?

- The fact that delivery mode is a key factor in strain transmission is very interesting but is not understood from the figures. The bacteria that are not significant between modes can be taken out of the graph (fig 3b). In addition the strains that were transmitted can be investigated more. Maybe which strains were transmitted in a certain species while others didn't?

Reviewer #2 (Remarks to the Author):

The manuscript by Feehily, O'Neill, Walsh, et al. presents an investigation of the transmission of Bifidobacterial species from mothers to infants using both cultivation and in-silico approaches. The work is based on 132 mother-infant pairs from the MicrobeMom study.

The manuscript is well-written and organized. One minor concern could be about the minimum number of reads of 100,000. What is the minimum number of reads for the metagenomes considered? Could this be a limitation in retrieving more MAGs?

One point about the delivery mode, will more species be vertically transmitted via breastfeeding (for example) later on? Could this be assessed using samples collected at the 3-month time point?

REVIEWER COMMENTS

Please note any line references mentioned here refer to those in the marked up document.

Reviewer #1 (Remarks to the Author):

Feehily et al. present an interesting manuscript examining the transmission of Bifidobacterium strains from mother-to-child using metagenomic and culturomic approaches. Their cohort, and the amount of isolated strains are very impressive and are surely in the great interest of the field. However, the story in this manuscript is not clear, and the analysis is not thorough enough, or maybe simply not explained. I hope the authors will take the suggested feedback and produce a revised manuscript that will enable them to highlight their unique cohort and isolates.

Thank you for taking the time to review our manuscript. We are grateful that you have found our study interesting. Following from your comments we have re-written several sections of the manuscript to better convey the message and provide a clear read.

Major comments:

- The paper reads like a list of small details, without a clear storyline. The text is jumping from one topic to another without any deep insights in each. A better description of the transferred strains, compared across delivery modes, perhaps even with a functional analysis, will be much more informative, and can teach the community about the factors that impact infant colonization in early life.

We have provided more clear aims in our introduction (line 86-91) and reordered certain results sections with more succinct sub-headings. Specifically, the results section has been reordered to follow a logical theme of metagenomic results-> genomic results-> evidence of strain sharing-> factors influencing sharing.

- There are many statistically significant p-values in the text, but it is unclear what they stand for, or how they were calculated. For example, "Parity was observed to be significantly associated with the vaginal microbiome ($p=0.0019$)". What does this mean? Is it associated with an increase of specific bacteria? Decrease of others? The reader cannot take away any knowledge from this. There is no mention of a figure showing this comparison, or analysis clearly, so the results section read as a list of dead-end streets.

Thank you, we have now included the statistical tool used to generate each p-value beside each occurrence. With respect to *envfit* analysis, this is a commonly used multiple regression method that determines if there are relationships between covariates and the beta diversity of the microbiota (in this case NMDS ordination). We have modified the text to clarify (line 139-153) and provided a more thorough description of the method in the methods section (line 572-583). The results of this analysis are shown in Fig S1 d and we have increased the size of this figure to improve readability.

- The figures are not readable. Not in their font size, nor in their attempt to convey a message. For example, in Figure 1: (a) it is unclear what the "|" mean in the numbers, also, which infants were born vaginally or by CS? (b) showing prevalence and abundance on the same plot is very confusing, two small panels are better than this combined view. (c) What are the different blocks of the phylogenetic tree? It is never explained. In Figure 2F the colors are never explained, they use the

same pallet of colors in f and g and as far as I understand, for two different reasons. Figure 3b is again re-using the same two colors only each time for a different variable. All these issues are making the figures really hard to interpret (with and without the accompanying text).

We have now increased the resolution of all figures and remade several panels throughout in line with comments below. In addition, we have provided more description in each figure legend to aid interpretation.

Specifically;

(a) We have remade Fig 1 (a) to clarify the difference in maternal and infant samples. The legend now clarifies that the “|” delimiter indicates maternal or infant samples. The breakdown of delivery mode and other statistics are presented in Table 1. We have edited the first two sentences of the results so that both this table and Fig 1 (a) are referenced together (Line 96-98)

(b) We thank the reviewer for raising this and we agree that this visual representation may be confusing and, thus, we have now split the plot into separate graphs for relative abundance and prevalence. The related plot from the previous version of Fig S1 has also been corrected and now included in the main panel.

(c) The blocks are split by timepoint before being clustered by Bray-Curtis dissimilarity. We have clarified this in the figure legend and moved this annotation bar closer to the dendrogram.

2(f) The colours here were included for initial aesthetic choice and do not have a relationship to similar colours in other panels. We appreciate that this could cause confusion and, thus, we have adjusted this plot to a single colour tone as the initial palette was unnecessary (now figure 3).

- The mother and infants have different bifidobacterium species, a discussion of this difference in light of transmission will be helpful.

It is clear that host diet is a major factor in the bifidobacterial composition. Species have a different repertoire of carbohydrate utilisation pathways that relate to what the host is eating. Despite the difference of species dominance between mothers and infants, transfer still occurred. This was mostly with species that are not detected with metagenomics alone due to the low relative abundance. We have included some discussion in relation to this point in the manuscript (line 475-482).

Minor comments:

- Throughout the manuscript the amount of reads per metagenomic sample is not mentioned. This is important since the authors claim that MAGs did not cover a lot of the bacteria present in the sample. Perhaps deeper sequencing would solve that.

We have now included the median number of high quality reads for each sample type at the respective relevant lines in the results section (line 102, line 112, line 123, line 127, and line 135). In addition, and also in agreement with the comment from Reviewer 2, we have included analysis and a figure relating to the relationship between sequencing reads and MAG recovery (line 207-209).

- In addition, in lines 149-151, it is mentioned that “372.50-fold genome coverage is a very high coverage”. With this coverage, we would expect significantly less than 26 contigs per genome. Are there a lot of outliers? What is the median coverage?

The median coverage was 359.32 (now added to the manuscript; line 179) and there were very few outliers as seen in Table S2. Higher depth does not always mean a better assembly, only that one can have greater confidence in identified SNPs. Excessive depth can actually harm a short read-only assembly by introducing low frequency sequencing errors that can confuse the algorithm. The main issue when assembling a genome from short read-only data is not depth or coverage but ambiguities in the genome assembly graph where the algorithm cannot confidently determine the correct placement of a contig, often due to repeat regions. The reference in the text (*Lugli, G. A. et al. Comparative genomic and phylogenomic analyses of the Bifidobacteriaceae family*) puts these genome assembly statistics in the context of a comparable *Bifidobacterium* genome dataset and shows that the contig number is as expected.

- Figure 2f: This is not a venn diagram as written in the caption. What do the bars represent? Not clear.

We apologise for this confusion; the legend was erroneously retained from a previous draft. We have now updated this to reflect that the panel displays an upset plot and presented this in the updated Fig 4.

- Figure 3b: same colors used for different things

As with the colour scheme in the previous Fig2 (as outlined above), we have adjusted the colour tone for Fig3. The updated version is now Figure 5

- Line 119 refers to fig1d, but the vaginal microbiome is not described there (to the best of my understanding)

Thank you for identifying this. Indeed, this was a typo and the text should have referred to Fig 1c (which relates to vaginal microbiota diversity). The correct reference is now in the manuscript.

- *B. breve*: why were these 9 strains sequenced, they seem to be the same on the phylogenetic tree? Why were any differences expected to be found? Once none were found why is this interesting?

Strains were selected for further sequencing to determine if there were any differences within the group *B. breve* (as we refer to them here) that could not be detected by short-read sequencing alone – such as the inversion event, conjugative element, and prophage regions reported. These specific strains were selected as they represent the full phylogenetic diversity of the group II clade.

- The fact that delivery mode is a key factor in strain transmission is very interesting but is not understood from the figures. The bacteria that are not significant between modes can be taken out of the graph (fig 3b). In addition the strains that were transmitted can be investigated more. Maybe which strains were transmitted in a certain species while others didn't?

Thank you, we have remade Fig 3a (now Fig 4a) to more clearly show the summary differences in shared strains between each category. We have also performed an analysis to determine what species are significantly different between them (Fig 4c). The text has been modified to include these changes (line 401-404).

Reviewer #2 (Remarks to the Author):

The manuscript by Feehily, O'Neill, Walsh, et al. presents an investigation of the transmission of Bifidobacterial species from mothers to infants using both cultivation and in-silico approaches. The work is based on 132 mother-infant pairs from the MicrobeMom study.

The manuscript is well-written and organized. One minor concern could be about the minimum number of reads of 100,000. What is the minimum number of reads for the metagenomes considered? Could this be a limitation in retrieving more MAGs?

In maternal stool samples, there were no MAGS recovered below a read depth of 1.4 million high quality reads. This compares to the vaginal samples (where diversity is much lower), from which we could recover MAGs with 120K high quality reads. We have now included some text relating to this finding in the results section (line 207-209) as well as a figure.

One point about the delivery mode, will more species be vertically transmitted via breastfeeding (for example) later on? Could this be assessed using samples collected at the 3-month time point?

Thank you, indeed this is an interesting point and we expect that strain sharing dynamics could change over a longer period. Unfortunately, it was not possible to investigate this in this study with the funds available.

REVIEWERS' COMMENTS

Reviewer #1 (Remarks to the Author):

The manuscript is in much better condition, and all of our concerns have been addressed. The re-ordering of the manuscript, especially with the new headers, is much easier to read and follow the story.

I have only a few small comments about technical aspects of figures/writing:

1. Figure 2 -

(a) Is the x-axis in log scale? If so, please indicate this.

(b) Something about the formulas is wrong. They don't match the lines, perhaps there is a difference between the scales, but the plot and the formula should be on the same scale. We assume the "c()" means constant? But it's not needed in this format here.

2. In general, I think that a figure describing the following sentence from the text could be more convincing for why culturomics is needed, instead of the current figure 2B. This is only a suggestion. "A total of 47/274 Bifidobacterium MAGs were found to have an average nucleotide identity (ANI) > 99.9 % and genome coverage > 90% to the 460 Bifidobacterium isolates. Taken together, the rather modest overlap between MAGs and cultivation-retrieved genomes suggests that the combined use of methods provides a more complete view of the Bifidobacterium strains present in samples."

3. In Figure4a, the green dots should be plotted on top of the gray ones, so they are visible when they overlap (like in the case of PX001, PX092, and so on).

4. When using the envfit function, you should reference the vegan package. I don't think that you need to mention this or each p-value (that is for the editor to decide), but the method itself is much clearer now that you have elaborated it in the text.

REVIEWERS' COMMENTS

Reviewer #1 (Remarks to the Author):

The manuscript is in much better condition, and all of our concerns have been addressed. The re-ordering of the manuscript, especially with the new headers, is much easier to read and follow the story.

I have only a few small comments about technical aspects of figures/writing:

1. Figure 2 -

(a) Is the x-axis in log scale? If so, please indicate this.

We have now detailed this in the figure legend.

(b) Something about the formulas is wrong. They don't match the lines, perhaps there is a difference between the scales, but the plot and the formula should be on the same scale. We assume the "c()" means constant? But it's not needed in this format here.

We have corrected this vertical line; it was a mix up between the log scales on the plot and the wrong formula was piped to the plot.

2. In general, I think that a figure describing the following sentence from the text could be more convincing for why culturomics is needed, instead of the current figure 2B. This is only a suggestion. "A total of 47/274 Bifidobacterium MAGs were found to have an average nucleotide identity (ANI) > 99.9 % and genome coverage > 90% to the 460 Bifidobacterium isolates. Taken together, the rather modest overlap between MAGs and cultivation-retrieved genomes suggests that the combined use of methods provides a more complete view of the Bifidobacterium strains present in samples."

We have visualised this information as a Venn Diagram (Fig 2 b) in addition to the other panels in Figure 2 as we believe that each show important distinct information. In making this figure we discovered that there were some erroneously included genomes that should not have been included. The correct number of Bif genomes is in fact 449. This difference does not influence any of the conclusions or main analysis and we have corrected summary figures (Fig 1 d & Supp Fig 1 f) and respective text (Line 156-160)

3. In Figure4a, the green dots should be plotted on top of the gray ones, so they are visible when they overlap (like in the case of PX001, PX092, and so on).

We have now corrected this.

4. When using the envfit function, you should reference the vegan package. I don't think that you need to mention this or each p-value (that is for the editor to decide), but the method itself is much clearer now that you have elaborated it in the text.

We have included a reference to *vegan* on lines 133 and 484, and have removed its name before each *p* value, but retained the *p* values themselves.